# Harnessing Ultrasound for Targeting Drug Delivery to the Brain and Breaching the Blood–Brain Tumour Barrier

**DOI:** 10.3390/pharmaceutics14102231

**Published:** 2022-10-19

**Authors:** Anita Barzegar-Fallah, Kushan Gandhi, Shakila B. Rizwan, Tania L. Slatter, John N. J. Reynolds

**Affiliations:** 1Department of Anatomy, School of Biomedical Sciences, University of Otago, Dunedin 9016, New Zealand; 2Brain Health Research Centre, University of Otago, Dunedin 9016, New Zealand; 3School of Pharmacy, University of Otago, Dunedin 9016, New Zealand; 4Department of Pathology, Dunedin School of Medicine, University of Otago, Dunedin 9016, New Zealand

**Keywords:** ultrasound, blood–brain barrier, ultrasound-responsive drug delivery, brain tumour

## Abstract

Despite significant advances in developing drugs to treat brain tumours, achieving therapeutic concentrations of the drug at the tumour site remains a major challenge due to the presence of the blood–brain barrier (BBB). Several strategies have evolved to enhance brain delivery of chemotherapeutic agents to treat tumours; however, most approaches have several limitations which hinder their clinical utility. Promising studies indicate that ultrasound can penetrate the skull to target specific brain regions and transiently open the BBB, safely and reversibly, with a high degree of spatial and temporal specificity. In this review, we initially describe the basics of therapeutic ultrasound, then detail ultrasound-based drug delivery strategies to the brain and the mechanisms by which ultrasound can improve brain tumour therapy. We review pre-clinical and clinical findings from ultrasound-mediated BBB opening and drug delivery studies and outline current therapeutic ultrasound devices and technologies designed for this purpose.

## 1. Introduction

Despite the growing body of literature in the development of novel and promising therapies, treatment of primary and metastatic brain tumours remains a major challenge worldwide [1,2]. In particular, limitations in the complete surgical removal of tumours, severe side effects of non-specific therapies, drug resistance, and importantly the blood–brain barrier (BBB), which is impermeable to most chemotherapeutic drugs, make treating brain tumours very complex [3].

Vascular permeability within a brain tumour (known as the blood–tumour barrier; BTB) is heterogeneous and is on average greater than the BBB. This results in penetration of the anti-cancer drug through the BTB and into tumour tissue. However, the increase in drug permeability is not enough to achieve an effective tumour therapy [4,5,6,7]. Several approaches have been developed to bypass the BBB/BTB; however, these approaches are either invasive, unable to increase localized permeability, insufficient at achieving adequate concentrations of delivered compounds to the brain tissue, or elicit systemic toxicity and potentially cause uncontrolled deleterious side effects in other organs [8,9,10]. Given the pitfalls in many of these approaches, a less-invasive strategy is warranted; a strategy to locally increase the permeability of the BTB to enable therapeutic concentrations of chemotherapeutic agents to reach the tumour site for improved efficacy, while minimizing off-target effects outside of the brain.

Over the past few decades, nanomedicine has gained significant interest in the field of controlled and targeted drug delivery systems (DDS) [11]. Nano/micro-particles (NPs) offer several advantages in the delivery of a variety of drug payloads, such as prolonging the circulation lifetime by controlling the pharmacokinetics of the drugs and hence accumulation at disease sites [12]. These properties have been largely beneficial in enhancing the safety and tolerability of cytotoxic drugs. However, the utility of these formulations in brain drug delivery has had limited success due to low penetration through the BBB or BTB. Given the beneficial properties of nano-formulations and immunological treatments for cancer and their limitation in brain drug delivery, there has been an increasing interest in the development of minimally invasive BTB disruption strategies to increase the penetrability of NPs to the tumour site.

Focused ultrasound (FUS) is currently the only available technique that can induce selective and localized opening of the BBB non-invasively in humans. The discovery of FUS-induced BBB opening has remarkably facilitated brain drug delivery [13], particularly in conjunction with nano/micro-formulations. Focused ultrasound not only facilitates therapeutic agent transfer across the BBB and BTB from loaded nano/micro-particles, but also accelerates the release of payloads from these particles [14]. Hence, FUS-responsive drug delivery using NPs is considered a promising approach in brain drug delivery, particularly in the delivery of loaded highly cytotoxic drugs for the treatment of brain cancers.

Here, we will provide a comprehensive review of the application of ultrasound in controlled drug delivery, from the basics of ultrasound and sono-responsive carriers to the clinical application of sophisticated ultrasound transducers for treating brain tumors. We will initially review the challenges in treating brain tumours, and strategies used to bypass the BBB/BTB to enhance drug delivery. Sono-responsive drug delivery, and therapeutic ultrasound devices developed for this purpose, will then be reviewed in detail. Finally, pre-clinical models and clinical studies using ultrasound-mediated BBB opening for the treatment of brain cancers will be discussed.

### 1.1. Therapeutic Challenges in Brain Cancer

Malignant gliomas, of which glioblastoma (GBM) is the most common primary brain cancer affecting adults, cause one of the highest levels of morbidity and mortality among cancer patients [15]. This mortality has changed minimally compared to other cancer types during the last three decades [16,17]. Current guidelines for brain cancer therapy include surgery, radiation, and cytotoxic chemotherapeutic agents, which have significant side effects and limited efficacy [18]. The median survival for GBM patients is 3 months following surgical resection only, improving to 8 and 15–18 months when radiotherapy [19] or radiotherapy plus temozolomide chemotherapy, respectively, are used as adjuncts to surgery [20,21]. Recently, an increase in the incidence of CNS tumours has been reported from multiple studies, possibly due to advances in the diagnosis of primary brain tumours, or in the treatment and improved survival from systemic cancer [22,23].

Multiple mechanisms contribute to treatment inefficacy. Firstly, complete surgical resection of brain cancers in many cases is nearly impossible due to their anatomical location and infiltrative nature. Secondly, the use of radiotherapy is often restricted, due to its potential deleterious effect on surrounding normal brain tissue. Thirdly, aggressive brain cancer cells show significant resistance to chemotherapy. Lastly, the BBB or BTB restricts the accessibility of malignant tissues to most chemotherapeutic drugs, thereby limiting the effective tissue concentration of agents to which malignant tissue might otherwise be sensitive [3].

The brain tissue landscape and its specific microenvironment is distinct to many other tissues [24,25], most prominently due to the presence of the BBB, which serves as the major barrier between the brain parenchyma and the circulatory system. Overcoming this barrier and increasing transport across it has been a major hurdle in the delivery of diagnostic and therapeutic compounds to the brain [26]. Formed by specialized endothelial cells firmly interconnected by tight junctions and further surrounded by pericytes, microglia, and astrocytic endfeet at the basal surface (Figure 1), the BBB prevents the entry of most blood-borne substances into the CNS [27]. Essential nutrients (e.g., oxygen, glucose), ions (e.g., sodium, potassium), small hydrophilic molecules, and other required substances by neurons are delivered across the BBB, via distinct transcellular processes such as passive diffusion and active transport. In addition, the diffusion across the BBB is generally only applicable for molecules of low molecular weight (MW < 400 Da) and high lipophilicity [22]. Furthermore, the BBB has a strong efflux transporting system made of highly expressed P-glycoprotein (P-gp) and multidrug-resistant protein efflux pumps that expel drugs out of the brain parenchyma.

The BTB is structurally highly heterogeneous and tortuous, with non-uniform and irregular permeability and active efflux of molecules, in comparison to normal BBB [4]. For perspective, increased transport of normally impermeable drug molecules, such as liposomal doxorubicin (580 Da) [28] and datasinab (488 Da) [29], have been reported across the BTB as compared to regions with an intact BBB. In the tumour core, leaky vessels elevate interstitial pressure and hinder convective fluid transport across vessel walls, preventing chemotherapeutic agents from reaching cancerous cells at therapeutic concentrations [30]. Furthermore, at the tumour periphery, the BBB remains relatively intact and therefore chemotherapeutic agents cannot easily enter, resulting in untreated malignant cells that often lead to tumour re-growth [31]. High-dose chemotherapy, as is still used in clinical protocols [32], can enhance the therapeutic efficacy in the tumour site, with the trade-off of increased risk of uncontrolled adverse systemic effects, thereby incurring high costs from off-target morbidity and drug usage [8,9]. To address these clinical concerns and achieve therapeutic drug concentration in cancerous brain tumours with minimal systemic distribution, techniques that allow local and less-invasive delivery of anti-cancer compounds to the CNS are required.

### 1.2. Nanomedicine and Limitations in Brain Drug Delivery

From a pharmaceutical perspective, NPs are typically defined as particles less than one micron in the longest axis that contain an active pharmaceutical ingredient [33]. The most commonly used NPs in drug delivery include liposomes, polymeric NPs, micelles, dendrimers, nanobubbles, phase-shift nanodroplets, and inorganic NPs made of iron oxide, gold, and quantum dots [11,34]. The benefits of NPs such as liposomes in the delivery of drugs have been well established [12]. In addition, the relatively large size of NPs (compared with free drug) means they can substantially prolong the circulation lifetime of drugs and target tumour tissue via enhanced permeability and retention (EPR) effects of macromolecules [12]. These properties have proven beneficial in enhancing the safety, tolerability, and efficacy of chemotherapeutic drugs such as doxorubicin and paclitaxel. This has best been exemplified by the reduced cardiotoxicity observed in patients administered liposomal doxorubicin when compared with free doxorubicin [35]. In fact, due to the improvements in patient morbidity following liposomal doxorubicin therapy, the United States Food and Drug Administration (FDA) approved NPs (Doxil^®^) for the treatment of Kaposi’s sarcoma in 1995 [36,37]. About 10 years later, the use of NP albumin-bound paclitaxel (Abraxane^®^) was approved for the treatment of a wide range of cancers including metastatic breast cancer [38,39]. Although improvements in patient safety and morbidity led to clinical approval of nanoparticle platforms, the efficacy of drug delivery in patients still remains modest, with only marginal improvements being observed relative to conventional formulations [37].

Whilst NPs improve the circulation half-life of conventional drugs and increase the likelihood of drug accumulation at the target tissue site, they face a wide range of biological barriers that significantly dampen their therapeutic efficacy. When the target tissue is CNS, the situation is even more complicated, as it is necessary to transport therapeutic agents across the BBB, the permeability of which is highly limited, particularly for large-molecule-like NPs, antibodies, recombinant proteins, or gene therapeutics [40]. Therefore, a major beneficial property of nano-formulations for brain delivery may be their ability to improve the bioavailability and circulation half-life of traditional drugs, without requiring the NPs themselves to cross the BBB/BTB [41,42].

### 1.3. Strategies to Enhance Drug Delivery across the BBB/BTB

A number of methods of bypassing the BBB/BTB to improve drug delivery locally to brain tumours have been trialed, such as direct intra-cranial injection [43], convection-enhanced delivery [44,45], and controlled release from polymer implants [46]. These have been found to improve drug concentrations at the targeted region of the brain, but with the cost of invasive open surgery, and significant risk of morbidity and mortality [47].

To date, several approaches to transiently increase BBB permeability, known as “opening” of the BBB, have been investigated. Osmotic or biochemical BBB disruption is performed by intra-arterial administration of hypertonic agents, such as mannitol, usually via the carotid or cerebral arteries. Although this has yielded moderate improvement in the delivery of anti-tumour drugs to the brain compared to systemic injection alone, the delivery is largely reversed within 10 min and not isolated to the tumour [48,49]. Co-treatment with vasodilatory drugs such as nitric oxide donors to improve the permeability of BBB has been found effective at increasing brain drug delivery [50,51]. These techniques, however, affect the entire volume of tissue supplied by the injected artery and increase intracranial pressure, leading to the diffusion of cytotoxic chemotherapeutic agents to normal brain tissue and the possibility of inducing seizures [1,52]. Overall, these strategies are accompanied by several limitations which hinder their clinical utility for safe and selective drug delivery to CNS. In particular, these strategies are either invasive or unable to increase localized permeability, or may result in insufficient improvement in the concentrations of delivered compounds, or induce significant brain toxicity.

The application of ultrasound is another strategy that has been utilised to transiently disrupt the BBB to improve drug delivery into the brain. Since its first description in the 1940s, significant improvements have been made in altering ultrasound parameters to reduce the potential of induced tissue damage, and in developing non- or minimally invasive delivery methods of ultrasound [53]. Given the relative safety, availability, low costs, and high efficacy of ultrasound technology, ultrasound-mediated BBB opening has become a hot topic in brain drug delivery [54]. Recent developments in ultrasound techniques such as MR-guided focused ultrasound (MRgFUS) and Transcranial Pulse Stimulation (TPS) have provided a unique toolbox to potentially overcome some of the limitations of brain cancer therapies.

The advantage of ultrasound application for drug delivery to particular tissues and disease sites is its ability to be focused tightly using lower operating frequencies, potentially deeply inside the body [55]. Drug delivery can benefit from three complementary effects of ultrasound application. Firstly, there is an increase in the blood supply to the targeted tissue via minor hyperthermia [56] (see Section 1.5.1 re thermal effects of ultrasound). Secondly, the permeability of the vascular endothelial barrier and cell membranes can be increased (both via mechanical and thermal mechanisms), and thirdly, drugs loaded in micro/nano-formulation carriers can be triggered to release locally into the target tissue [55].

Despite the relatively leaky vasculature in some regions of the tumour [57], further enhancement of drug extravasation has been sought as an effective strategy for tumour targeting, particularly when only a very small fraction of the intravenously administered NPs will have a chance to reach tumour tissue, and an even lower amount will be deposited there [55]. Therefore, the combination of ultrasound with drugs and drug carriers has the ability to improve drug delivery to the target tissue in the CNS. Following insonation, the endothelial barrier may stay open for minutes to days, improving the potential for intravascular circulating drugs or drug carriers to exit the bloodstream and accumulate in CNS tissue [55]. The advantage of ultrasound application for drug delivery to particular tissues and disease sites is its ability to be tightly focused, and using lower operating frequencies, potentially deeply inside the body [55]. Given the deep penetration of ultrasound, this transient hyperpermeability can be used to increase drug delivery in any tissue [58]. This therapeutic strategy is particularly relevant in brain cancer therapy, and has recently translated into clinical trials involving brain malignancies, as well as in other CNS pathology [59].

### 1.4. Physics of Ultrasound

Ultrasound is composed of sound waves that exceed the audible range of human frequencies (>20 kHz, Figure 2), and function as a pressure wave through a physical medium (e.g., air or water). Ultrasound waves are usually generated via a piezoelectric transducer, which converts electrical energy into mechanical movement [60]. In other words, ultrasound waves constitute the oscillatory movement of molecules (about fixed points) in the medium at high and low pressure, corresponding to compression and rarefaction, respectively [60,61]. It is these mechanical mechanisms through which ultrasound is theorized to affect biomolecules and disrupt biological barriers. In comparison with light, ultrasound energy is relatively less absorbed within water and many soft tissues, thus allowing it to penetrate deeply into the body and transmit energy to a precise location within target tissue [62]. Such properties of ultrasound are used for the design of diagnostic ultrasound scanners and therapeutic devices that are currently in clinical use.

### 1.5. Historical Perspective of Diagnostic and Therapeutic Applications of Ultrasound in Brain

Although ultrasound was discovered around 12 years before the X-ray (1883), its biomedical applications were found much later. Since the 1940s, ultrasound has been clinically used as a form of non-ionizing radiation energy in medical diagnostics and image-guided interventions [9]. Within the range between 0.8 and 20 MHz [63], ultrasound has been frequently used in bioimaging, physical therapy [64], hyperthermic cancer therapy [65,66], and more recently for controlled drug delivery.

In contrast to diagnostic use of ultrasound (ultrasonography), its therapeutic applications are exhibited through the use of comparatively lower frequencies (0.4–2 MHz) and higher intensity parameters. This enhances the deposition of energy amongst sonicated tissue, inducing the desired therapeutic effects, which can range from mild, as used in ultrasound-mediated drug delivery, to the more extreme, as used for thermoablative purposes [67,68].

Potential therapeutic effects of ultrasound were initially studied for selective tissue ablation in 1942 by applying high-frequency and short-wavelength ultrasound to induce specific lesions on the cortical and subcortical areas of the brain [69]. With the advancement of technology in the 1950s, transducers with the ability to target deeper brain structures associated with psychiatric conditions were developed [70,71,72]. These early investigations laid the foundation for the subsequent development of sophisticated FUS surgical tools for brain cancers [73,74,75,76,77,78]. However, early attempts in brain tumour therapy using FUS were conducted intraoperatively, via a craniotomy through which the sonication was performed, prior to replacing the skull flap at the end of the procedure. Over the past two decades, the development of new ultrasound technologies and the availability of real-time imaging techniques have propelled the therapeutic application of ultrasound exponentially.

The implementation of phased array transducers and real-time MRI thermometry monitoring has made transcranial FUS (i.e., ultrasound into the brain without the need to open the skull) feasible. In this method, with the specific information collected from the patient (e.g., head CT scans or acoustic measurements) and individual adjustments to the phase and amplitude of transducer elements, it became possible to correct the aberrations caused by the skull during ultrasound propagation [79,80]. In addition, by coupling with MRI, the formation of a lesion during sonication can be closely monitored [81,82].

With small craniotomies and direct placement of small and bio-compatible transducers on the dural surface, most skull-related aberrations can be bypassed [83]. Furthermore, for some low-power applications, a patient-specific acoustic lens can be utilized as a cost-effective strategy to focus the ultrasound beam generated from the transducer (a process known as lens-based aberration correction) [84]. In addition to these clinical successes, ultrasound has been employed for the treatment of many other conditions involving other tissues [85,86].

Given the large range of ultrasound therapeutic applications, there has been an increasing interest in the potential mechanisms by which ultrasound acoustic energy interacts with cells, tissues, and therapeutic agents or carriers [87]. These mechanisms are typically divided into thermal (heat generation) and non-thermal mechanisms (acoustic cavitation and acoustic radiation forces) (Figure 3) [9,88].

#### 1.5.1. Thermal Effects of Ultrasound

Depending on the parameters of ultrasound employed, within exposed tissues a given proportion of acoustic energy will be transformed into thermal energy, thus exerting thermal effects (Figure 3). The rate of temperature increase in the targeted tissue depends on both tissue properties (e.g., the tissue density and distance from the ultrasound source) and the ultrasound exposure parameters (e.g., frequency, intensity, pulsed delivery, exposure duration) [9]. Given the importance of temperature in biomedical reactions, enzymatic activities and immune system responses, the temperature rise at a specific tissue location can affect its physiological function. It has been shown that the cellular and physiological adverse effects of mild, transient hyperthermia (below 40 °C) are generally insignificant. However, prolonged hyperthermic changes (>40 °C) yield irreversible conformational changes in vital proteins and cellular structures [89].

**Figure 3 pharmaceutics-14-02231-f003:**
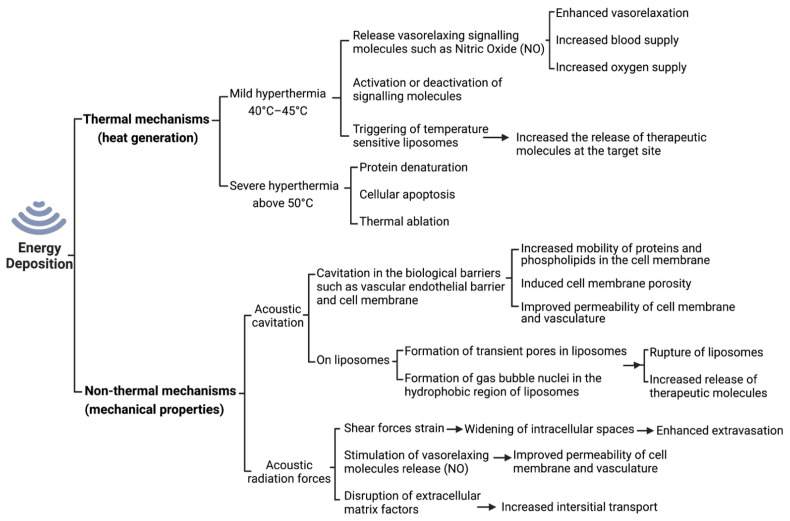
A schematic representation of the manner by which energy deposition from ultrasound exposures can act through various mechanisms. When ultrasound waves are applied to the targeted tissues, they produce mechanical, thermal, chemical and biological effects. The biological effects of ultrasound depend on the applied intensity, and depending on the dose and duration of administration, ultrasound can produce reversible or irreversible effects [90]. Adapted with permission of Elsevier from ref [88]; permission conveyed through Copyright Clearance Center, Inc.

Mild, ultrasound-assisted hyperthermia (40 °C to 45 °C) has been explored for its therapeutic potential in the treatment of many solid tumours, including brain cancers [91,92]. Given that most solid tumours are irregularly vascularized and the tumour microenvironment is hypoxic and acidic, tumour cells can tolerate elevated heat-induced stress better than healthy normal cells. Nevertheless, mild heating of tumour tissue was found to be beneficial for radio- and chemo-sensitization and controlled drug delivery in tumours [93]. Short-time FUS hyperthermia was able to significantly enhance the delivery of chemotherapeutic agent, liposomal doxorubicin, to cancerous tissues in the mouse brain, without affecting uptake in normal, healthy brain tissues [94]. Hyperthermia in tumour tissue also increases oxygen supply and deactivates proteins that are responsible for restoring damaged DNA [9].

Inducing high temperatures above 50 °C in target cells results in rapid protein denaturation and subsequent cellular apoptosis, which is maintained over seconds to minutes. This phenomenon is thermal ablation, which can be achieved via the use of high-intensity FUS (HIFU, e.g., 1000 W/cm^2^ intensity, estimated 4 MPa peak negative pressure at the focus in water, and 0.5 to 7 MHz frequency). Thermal ablation has applications for minimally invasive surgery of uterine fibroid and other tumour masses (e.g., brain, breast, liver, bone, and prostate) [95].

#### 1.5.2. Non-Thermal Effects of Ultrasound

The therapeutic effects of ultrasound can also occur via non-thermal mechanisms, in which radiation force and acoustic cavitation affect the biological activity of tissue (Figure 3) [96,97,98,99]. Non-thermal acoustic shock waves have been used to break up kidney and bladder stones [100], as well as sono-thrombolysis to recanalize acute intracranial arterial occlusion [101,102].

Ultrasound has been shown to improve the permeability of biological barriers and cavitation is considered the most important non-thermal mechanism for this [88,103]. Acoustic cavitation is the formation and or growth, oscillation, and collapse of small gas bubbles under the influence of the varying pressure field of a sound wave in a medium [103]. The resulting cavitation increases the mobility of proteins and phospholipids in cell membranes, cell membrane porosity, and intercellular gap formation, thereby altering the concentration of active molecules in the tissue environment. This effect has special value for CNS drug delivery due to the limited permeability of BBB to most compounds [104,105,106].

Of the physiological effects of ultrasound, we now discuss local vasorelaxation by releasing signalling molecules, cell membrane and vascular permeability.

#### 1.5.3. Ultrasound-Induced Vasorelaxation

Focused ultrasound has been shown to induce changes in blood pressure in a precise location, at least in part, through stimulation of endothelial cells to release nitric oxide (NO) and other vasorelaxing signalling molecules such as prostacyclin. Maruo et al. [107] showed that 55.5 kHz FUS, at an intensity of 50 mW/cm^2^ (acoustic pressure approximately 40 kPa) for 3 s, induced significant vasorelaxation in isolated canine arterial segments, with and without intact endothelium (precontracted with norepinephrine). They also found that early vasorelaxation (1 min after stimulus) was maximally inhibited by NO synthase inhibitor compounds such as N-Nitro-l-arginine (l-NNA). However, late vasorelaxation (5 min after the stimulus) was almost abolished via a combination of l-NNA and the prostaglandin release inhibitor, indomethacin, demonstrating a relative increase in prostacyclin activity and reduction in NO activity. No significant changes in the tissue temperature or disruption of endothelial cell integrity were observed in the study. Hence, the authors concluded that FUS induces vasorelaxation predominately through a time-dependent endothelial NO and prostacyclin release process, which appears unrelated to tissue heating or endothelial architectural disruption.

Ultrasound-induced vasorelaxation in humans was first reported by Lida et al. in 2006 [108]. They showed non-invasive transcutaneous low-frequency ultrasound dilates human brachial arteries with a rapid onset (2 min), lasting about 20 min. It is worth mentioning that the thermal effects are minimal when pulsed FUS is used, however, local hyperthermia will also cause localized vasodilation through the generation of a range of signalling molecules including NO [108]. Ultrasound-induced hyperthermia increases blood flow as well as blood vessel permeability.

#### 1.5.4. Ultrasound-Induced Permeability

The effects of ultrasound on the cell membrane and vascular permeability have been studied for several decades. The initial observation of ultrasound-induced BBB opening was found as a result of tissue damage in the early 1950s [109,110]. However, nearly 40 years later, this phenomenon was suggested as advantageous for the delivery of chemotherapeutic agents into brain tumours [111]. In 1995, Vykhodtseva et al. reported that ultrasound can reversibly improve BBB permeability in rabbit brains in vivo without significant damage of the BBB or brain parenchyma [112]. It is now well established that ultrasound can increase the permeability of cell membrane and vasculature through several mechanisms which will be discussed at cellular and tissue levels.

At the tissue level, similar to the vasodilatory effects (mentioned above), ultrasound can stimulate the release of several signalling molecules (e.g., NO and prostaglandins) which increase the permeability of blood vessels. NO decreases the recruitment of the Rac guanine-nucleotide-exchange factor (GEF) TIAM1, adhesion junctions and VE-cadherin (also known as cadherin 5), as well as reducing stress fibre formation. In addition, through Rho GTPase-dependent regulation of cytoskeletal architecture, NO leads to a reversible increase in vascular permeability [113].

Moreover, ultrasound can cause acoustic cavitation in endothelial cells, causing intercellular gap formation that increases permeability to the circulating therapeutic agents. In a complex medium such as tissue, the exact nature of the cavitation phenomenon is still not fully understood. It has been suggested that, upon sonication, the volume of oscillatory gas bubbles available in physiological fluids play a major role in the cavitation process. To amplify this phenomenon, gas-filled microbubbles (MB) can be added to the circulation. During ultrasound insonation, MBs oscillate by expanding at low pressure and contracting at high pressure. Depending on the pressure amplitude, the cavitation can be stable or inertial. In stable cavitation, the pressure tension induced by the external acoustic field can be tolerated by the MBs and the moderate volume oscillations result in micro-streaming of fluid around the bubble [114]. However, in inertial cavitation, acoustic pressure amplitude is sufficiently high and MB volume oscillations result in a net expansion that causes subsequent bursting. Such transient and violent collapse causes various destructive mechanical changes in the endothelial layer leading to increased vascular permeability (Figure 4) [115]. It is generally believed that inertial cavitation events result in adverse, and potentially permanent, structural alterations to the BBB [88]. This is evident by the use of passive cavitation detectors or acoustics emission monitoring that highlight reflective acoustic patterns from MB activity corresponding with inertial cavitation, e.g., broad-/wide-band emissions. These patterns of activity correspond to whenever tissues are undergoing damage as a result of ultrasound–BBB disruption [116]. In order to reversibly increase vascular permeability while minimising tissue damage, ultrasound parameters need to be set to prioritise the induction of stable over inertial cavitation.

At the cellular level, in addition to the cavitation facilitated by MBs, ultrasound increases the motility of phospholipids in the cell membrane, leading to a higher permeability. Additionally, it has been shown in pre-clinical studies that FUS combined with MBs can cause disassembling of the tight junction proteins between cerebrovascular endothelial cells (Figure 5) [117] and down-regulation of P-gp drug efflux pumps expression along the endothelial cell membrane [118,119,120]. While P-gp has been assessed in preclinical studies, further studies on focused ultrasound and its effects on different BBB transporters, such as breast cancer resistance protein (BCRP), are needed for a more complete understanding of such mechanisms. Acoustic cavitation of MBs has been recognized to play an important role in cell sonoporation, the formation of temporary pores in the cell membrane induced by ultrasound (Figure 5). Sonoporation has shown great potential as a non-viral strategy for drug and gene delivery via transient disruption of cell membrane [121,122,123,124]. The increased mechanical movement of molecules and hyperthermia at the site caused by the ultrasound also facilitates the passive accumulation of the drug/gene carriers into cells [125] (Figure 5).

**Figure 4 pharmaceutics-14-02231-f004:**
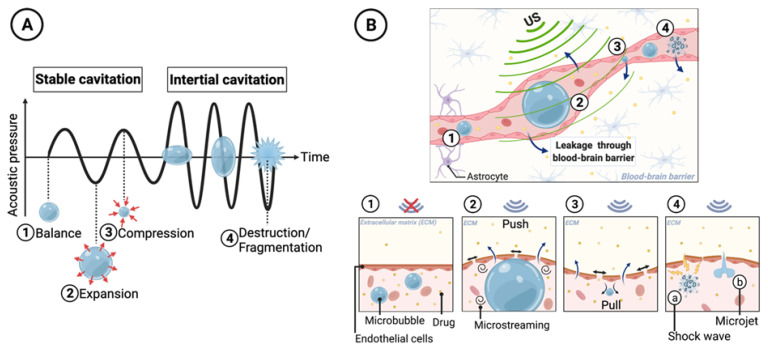
Mechanisms of therapeutic agent extravasation by conformational changes in blood vessels during microbubble (MB) cavitation, including transient disruption through stable cavitation and destructive opening through inertial cavitation. (**A**) The MB diameter changes when exposed to the pressure waves of ultrasound. At lower ultrasound intensities, MBs may undergo slight oscillations in size (stable cavitation). During the compression portion of the wave, the MB diameter shrinks, while during rarefaction the MB expands. At higher ultrasound intensities, the MB undergoes an unstable growth (inertial cavitation) followed by a rapid collapse and implosion. (**B**) Impact of ultrasound-induced cavitation on cell membrane permeability. (1) represents the vascular endothelium before ultrasound exposure as a significant barrier to the extravasation of circulating therapeutic agents. (2) shows MB expansion and microstreaming which lead to a temporary increase in the gap–junction distance between vascular endothelial cells, thus allowing circulating agents to extravasate. (3) represents MB compression. (4) MB bursting induced by inertial cavitation, resulting in microjets and shockwaves in the surrounding blood that can fragment nearby lipid membranes. Permission for adaptation of Panel A from [126] granted under Creative Commons Attribution 4.0 License, (http://creativecommons.org/licenses/by/4.0/; accessed on 15 August 2022), otherwise created with BioRender.com.

**Figure 5 pharmaceutics-14-02231-f005:**
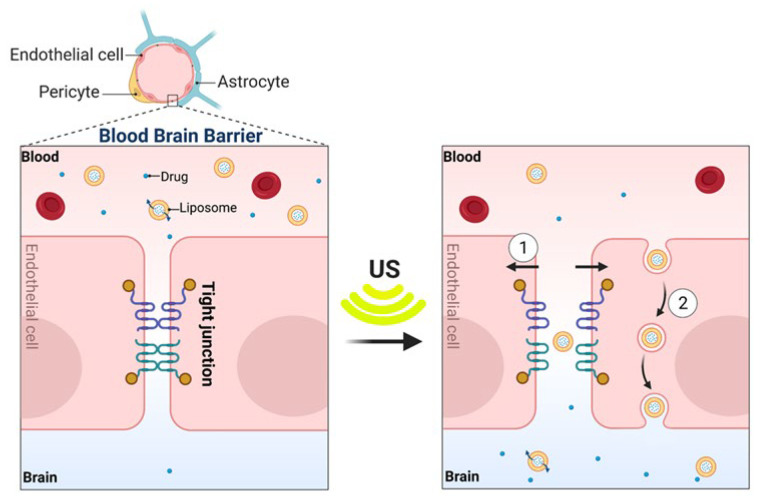
Mechanisms by which ultrasound affects the blood–brain barrier at the cellular level. (**1**) Interaction of MBs and ultrasound transiently disrupts the tight junction of the endothelial cells to facilitate paracellular transport. (**2**) Ultrasound-mediated mechanical forces increase the transcellular transport across endothelial cells by several membrane poration mechanisms to shuttle the drug out of the vessel lumen into the brain. Permission for adaptation from [127] granted under Creative Commons Attribution 4.0 International License, (http://creativecommons.org/licenses/by/4.0/; accessed on 15 August 2022), otherwise created with BioRender.com.

Numerous evaluations into the therapeutic potential of ultrasound-induced BBB permeabilisation for improving chemotherapeutic delivery in managing cancers have been conducted [9]. Synergistic cytotoxic effects were observed in many in vivo and in vitro studies when ultrasound was added to the chemotherapeutic drugs (reviewed by Pitt. WG. et al. [125]). For example, Loverock et al. have shown that 1 h sonication (2.3 W/cm^2^ at 2.6 MHz) of Chinese hamster lung fibroblast cells did not significantly impact cell viability. However, when the sonication was combined with doxorubicin treatment, it resulted in a significant enhancement in the cytotoxic activity of doxorubicin [128]. Tachibana et al. reported that a combination of low-intensity ultrasound (0.3 W/cm^2^ at 48 kHz) for 120 s with cytosine arabinoside (Cytarabine) lead to a 100-fold reduction in the number of colonies formed by human leukaemia (HL-60) cells [129]. Scanning electron microscopy of the insonated cells revealed that ultrasound altered the cell membrane, thus resulting in the increase of Cytarabine uptake into cells.

Despite the promising synergistic anticancer effects of ultrasound and chemotherapeutic drugs, systemic administration of chemotherapy is associated with significant off-target effects that cannot be totally alleviated by ultrasound. Hence, in recent years, research has focused on developing nanoformulations that can sequester chemotherapeutics inside a biological package, thereby minimising systemic drug effects and allowing more targeted uptake or release via ultrasound stimulation at the tumour site [125].

### 1.6. Ultrasound-Responsive Drug Release Systems

One of the main challenges in the delivery of cytotoxic anticancer drugs is the ability to control drug release kinetics at the target site. Several types of triggers have been used for the development of stimuli-responsive drug delivery systems, including pH [130,131,132], temperature [133], redox state of cancer cells [134], and enzymes [135,136,137,138]. Another approach is the use of external stimuli by which to control both the drug release location and profile, such as light, x-ray, magnetic field, and ultrasound [14,88,139,140,141].

Given that ultrasound is a generally well-tolerated, non-radioactive, accessible and less invasive method of energy transmission into the body, sono-responsive drug delivery systems have been the subject of much research in controlled and selective drug delivery [14,142,143]. Drug carrier systems can be formulated to be sensitive to the mechanical and/or thermal biomechanisms of ultrasound, thereby allowing drug release from drug carriers in the selected region of interest on ultrasound stimulus exposure. While other energy sources (e.g., magnetic field and light [144]) can also produce such changes, ultrasound can effectively focus energy deep within the body, facilitating local treatment of a broader range of conditions than alternative approaches. Similar to the mechanisms by which ultrasound increases membrane permeability, sono-responsive delivery systems can be designed to respond to the elevation in temperature and/or the mechanical effects of ultrasound waves. However, formulating carriers that can stably retain their payload during circulation and yet exclusively release it upon external ultrasound stimulation remains a major challenge.

#### 1.6.1. Thermal Effects of Ultrasound on Increased Drug Release from Nano/Micro-Particles

As previously mentioned, ultrasound can selectively increase target tissue temperature to several degrees above physiological temperature, rendering it suitable for controlled delivery of drugs. Although high temperatures can affect the pharmacokinetics of drug carriers in tissues, temperature elevations (mainly induced by ultrasound) can also induce the release of drug from the carrier.

Given that high temperatures can result in non-specific lesions or cause side effects, thermosensitive carriers are often developed to deliver their therapeutic content at a few degrees above physiological temperature (42–43 °C). These carriers can release up to 80% of their payload after 15 min of local hyperthermia at 43 °C [145]. It is very crucial that the formulations stay stable at physiological temperature and release their payload with mild ultrasound-induced hyperthermia in a sharp, fast, and quantifiable manner [146].

For example, in thermo-responsive liposomes, when the local tissue temperature increases using an external source of heat, such as infrared laser [147] or ultrasound [148], beyond the phase transition temperature of the lipid, it results in disruption of the close and ordered packing of the lipid bilayer. This introduces “free volumes”, which enable the payload to be released from the particles [133,149,150]. The first example of thermosensitive liposomes was published by Yatvin and Weinstestein in the late 1970s [151]. Liposomes were formulated with a specific phospholipid–di-palmitoyl-phosphatidyl-choline (DPPC), which resulted in a gel-to-liquid crystalline phase transition temperature of 41 °C. Hence, at 41 °C or above, the permeability of the bilayer was significantly increased, which allowed the release of the hydrophilic solutes encapsulated in the inner aqueous core of the liposomes. Since then, several thermo-responsive liposomes have been developed containing different drugs, including doxorubicin [152], cisplatin [153], methotrexate [154], and paclitaxel [155]. In addition, hyperthermia has been shown to increase the accumulation of liposomal doxorubicin at brain metastatic sites in mice [94].

Nevertheless, so far, only a small number of pre-clinical and clinical studies have reported the use of HIFU-induced local hyperthermia, in combination with thermosensitive drug carriers. ThermoDoxR^©^ is a thermo-responsive doxorubicin-loaded liposome with a phase transition temperature of 39 to 41 °C, being investigated in combination with ultrasound ablation in Phase III clinical trials, and in combination with HIFU treatment in Phase I trials [156,157].

#### 1.6.2. Mechanical Effects of Ultrasound on Increased Drug Release from Nano/micro-particles

Besides the thermal effect, ultrasound can also trigger drug release from drug carriers by inducing high mechanical stresses. This mechanism is more prevalent with the application of low intensity (≤5 W/cm^2^) ultrasound, intermediate ultrasound frequencies (1 to 3 MHz), and smaller duty cycles [157].

Liposomes are the most studied nanocarriers for sono-responsive drug delivery, particularly using unfocused stimulation. It is well established that low-frequency ultrasound (LFUS) increases the release of payload from liposomes [158,159,160,161], without affecting the drug’s chemical integrity or biological potency [159].

Investigation of the exact mechanical mechanisms through which the liposome bilayer allows the release of its drug payload is still ongoing. However, it is likely that the release can be achieved via the formation of transient pores and the subsequent rupture of liposomes [161], mimicking what occurs in the cavitation of cell membranes following ultrasound sonication. The oscillating ultrasound field initially forms gas bubble nuclei in the hydrophobic region of the sono-responsive liposome’s lipid bilayer. The subsequent mechanical stress then results in the collapse of nuclei and the generation of large transient pores across the membrane, through which the payload can escape from the particles.

It is of note that most self-assembled formulations are not rigid and the constituents of the particles, often lipids, are dynamic and constantly moving. Drug sequestrated into such a formulation (as a result of the concentration gradient during encapsulation) can slowly penetrate through the particles and release into the medium. Mechanical stimulation induced by ultrasound can speed up this phenomenon and accelerate the release profile [162].

#### 1.6.3. Sono-Responsive Carriers

As schematically shown in Figure 6, a wide range of ultrasound-responsive carriers have been designed with different materials and include liposomes (100–400 nm), micelles (50 nm), MBs (1–2 µm), silica nanoparticles (20–900 nm), droplets (1.5–4 µm), and nanobubble–nanoparticle complexes. Generally, sono-responsive carriers release their payload at a significantly higher rate in the presence of ultrasound.

Microbubbles are small gas-filled colloidal particles designed with a more or less flexible shell of lipid monolayer stabilizing the perfluorocarbon (PFC)/air gas core [163]. They were initially developed as a contrast agent for ultrasound imaging and diagnostics [164]. Currently, MBs are believed to have great potential as carriers for therapeutic substances such as drugs, small molecules, nucleic acids, and proteins [165,166]. Given that MBs can be expanded, imploded, or fragmented under sufficient acoustic pressures [164], the insonation of these agents requires additional safety consideration. However, it opens up new opportunities for drug delivery using ultrasound [166].

Micelles and liposomes are the most frequently used carriers among the various drug delivery systems used for targeted therapy [167]. Micelles are made of hydrophilic–hydrophobic interactions of molecules that self-assemble in aqueous solutions. Several sono-responsive micellar particles have been used for the delivery of chemotherapeutic drugs in cancer [14,168]. The topic of ultrasound activation of micelles has recently been reviewed [14,125,168,169].

The field of sono-responsive drug delivery has been strongly influenced by the development of liposomes as the main drug carriers [170]. Unlike micelles, liposomes are vesicles made of lipid bilayers in an aqueous solution which can encapsulate both hydrophilic and hydrophobic drugs in their aqueous phase and outer lipid bilayer membrane, respectively [125,171]. Since the early 1960s, there has been a lot of interest in the potential applications of liposomes for drug delivery, particularly after the advance of adding lipid-anchored hydrophilic polymers to improve stability and prolong circulation half-life [172,173,174,175]. The liposome surface is typically modified with polyethylene glycol (PEG) to prolong its clearance time [176].

Maeda et al. have reported that 40 kDa to 250 kDa macromolecules or particles incapable of experiencing renal filtration (diameter >5 nm), such as liposomal and micellar NPs, can accumulate in most tumour tissues. This is due to the pathological conditions within solid tumours that are not observed in normal tissues, such as defective vascular architecture and impaired lymphatic drainage/recovery system. This phenomenon, as aforementioned, is known as the EPR effects of macromolecules [177].

In addition, liposomal chemotherapeutic formulations have reduced systemic cytotoxicity when compared to free drug, an example being liposomal doxorubicin, which has reduced cardiotoxic effects in clinical studies [178]. Currently, numerous liposomal particles have been tested in pre-clinical animal studies, several in clinical trials of varying stages for cancer therapy, and a few available on the commercial market [179]. Doxil^®^, a PEGylated liposome loaded with one of the most commonly used anti-cancer drugs (doxorubicin), is known as the first liposomal nanomedicine approved by the Food and Drug Administration (FDA) in 1995 [36,180]. Since Doxil^®^, more than 10 other liposomal drugs with various sizes, structures, and lipid compositions with different therapeutic aims have been approved for clinical uses by the FDA, including Myocet^®^ (non-PEGylated liposome carrying doxorubicin) and DaunoXome^®^ (daunorubicin-loaded liposome) [162,177]. The currently available liposomal drugs in the United States are mainly antifungal and anticancer therapies; many more products, including those used as analgesics, gene therapies, and vaccines, are being developed [181]. Of note is the use of liposomal formulations in the two recently developed mRNA COVID-19 vaccines [182,183].

Using ultrasound as a stimulus, several factors influence the release of the drug, including ultrasound parameters (frequency, intensity, pulsed delivery and duration of application), the position of the ultrasound source, and the nanocarrier composition [184,185]. In the case of liposomes, the lipid composition and physical characteristics play a prominent role in the sensitivity to the ultrasound [157,185]. The membrane composition can be chemically modified to increase the sensitivity to ultrasound, as comprehensively discussed by Schroeder, A. et al. [14]. For example, the presence of amphiphiles, such as phospholipids with unsaturated acyl chains, increases liposome susceptibility to LFUS through destabilizing the lipid bilayer. Additionally, responsivity of liposome to ultrasound is greatly affected by introducing PEG-lipopolymers to the bilayer, most likely due to absorption of ultrasonic energy by the highly hydrated PEG head groups [14]. As discussed in Section 1.7, ultrasound-induced drug release from liposomes can occur through thermal and/or mechanical mechanisms, and are highly dependent on the physiochemical properties of the lipid bilayer.

Unlike thermal stimulation, the liposome shape and composition of the inner aqueous core affect drug release induced by mechanical stimulation [157]. In addition, some compositions, such as metal nanostructures (particularly hollow metal nanostructures such as hollow gold nanoparticles) are highly advantageous for the development of sono-responsive drug delivery. Hence, the combination of metal nanostructures with liposomes can unexpectedly enhance acoustically activated drug release in liposomes [143], for reasons that are not clear. Attachment of gold NPs or other metal nanostructures in liposomes is variable and dependent on the hydrophobicity and surface charge of the metal nanostructure [186]. These structures can be formulated within the liposomal core [187], tethered to the membrane [143,188,189,190], inserted within the bilayer [191], free in liposome solution [188,192], or assembled as aggregates with liposomes [193].

Another development in sono-responsive carriers has been the conjugation of drug-loaded particles, such as liposomes, onto the surface of MBs (Figure 6). This carrier conformation allows for additional mechanical sensitisation, as the insonation of the MB causes it to oscillate and circulate fluid around itself [194]. As the attached drug-loaded particles have a higher density than the surrounding fluid, they get sucked into the fluid flow surrounding the oscillating MB, thus experience sufficient shear stress, causing rupture and release of their drug payload [125].

In recent years, the emerging phase-changing ultrasound-responsive nanodroplets have attracted substantial attention for imaging and tumour therapy. Ultrasound-responsive phase change in the structure of NPs occurs through a process known as acoustic droplet vaporization (ADV). Generally, these NPs are made of a PFC core and a shell coating. The acoustic excitation of such NPs results in vaporization of the core into gas bubbles and a build up of mechanical pressure for sonoporation of cell membranes [195]. Phase shift droplets offer many advantages over micro/nano-bubbles and liposomes due to their higher stability and possibility of smaller sizes [196]. Micro/nano-bubbles and liposomes also suffer from limitations such as low spatial selectivity and short circulation time in vivo [195,197].

### 1.7. Pre-Clinical Models for Ultrasound-Mediated BBB Opening

The majority of research into therapeutic applications of ultrasound in brain cancer pharmacotherapy has focused on using ultrasound to reversibly modulate BBB opening to deliver normally impenetrable therapeutic agents to the brain. The magnitude and spatial extent of BBB opening, as well as the safety of this approach, depends on several ultrasound parameters, including transducer type, pressure amplitude, sonication time, transducer frequency, geometry of the transducer and number of sonication points, pulse characteristics (pulse length and pulse repetition frequency) as well as MB-related parameters [163,198,199]. A recent systematic review of such parameters and their effects on the extent and safety of BBB opening across pre-clinical and clinical studies was conducted by Gandhi et al. [54]. Using appropriate parameters, therapeutic agents ranging from low-molecular-weight drugs [59,200,201,202], larger molecular weight molecules such as monoclonal antibodies [139,203], gene vectors (both non-viral and viral) [204,205,206] and nanoparticles to even stem cells [207] and natural killer cells [208], have been successfully delivered to brain cancer sites in pre-clinical studies. The emerging evidence from these studies provides promising therapeutic potential in advancing the pharmacological management of brain cancers. Therapeutic compounds up to 2000 kDa in size can cross the ultrasound-induced BBB opening, depending mainly on the peak negative pressure of the ultrasound pulses [209]. However, at high enough pressures to allow the passage of molecules or cells larger than 500 kDa, BBB disruption and opening would result in microhaemorrhage [207,209]. Nevertheless, no statistically significant correlation between the ultrasound-mediated BBB opening and the resulting extravasation of red blood cells and long-term neural damage was detected [210,211].

Ultrasound-mediated BBB opening has been studied in multiple species with a wide range of skull volumes and thickness, from small rodent models and large animal models (e.g., swine, canine, sheep, nonhuman primates) to human participants, with targets in different parts of the brain and tumour tissue [212]. A variety of pre-clinical brain cancer models have been tested, and all have been rodent models with orthotopically implanted cancerous cells, derived from a variety of allogenic or xenogeneic cell lines [202]. Currently, no large animal neuro-oncological model exists, posing a potential limitation in the translation of pre-clinical findings to clinical research. Another translational limitation posed by widely adopted rodent neuro-oncological models is the higher permeability BTB that forms, as compared to the BTB formed in human GBM patients [213].

In some of these animal studies, it has been reported that the FUS-induced BBB opening occurs immediately upon insonation and resolves within hours to days following sonication, without inducing significant brain injuries [104]. Proof of concept that ultrasound can increase the BBB permeability has been well-established in several rodent models [212,214]. Additionally, in most experimental set-ups of ultrasound-induced BBB opening, a single transducer has been used as the source of ultrasound. In small animal models such as mice, a single transducer can stimulate a significant volume of the brain, accompanied by relatively low spatial specificity. This inevitably leads to hemispheric BBB disruption [212]. Furthermore, scaling from small to large animals in ultrasound-induced BBB opening can be quite challenging, particularly because the ultrasound beam is greatly affected by the skull thickness [215]. In this context, several other factors including brain mass/body weight, body size, and source (non-laboratory bred) have also been considered important to model the clinical setting [198].

Ultrasound-induced BBB opening has been successfully applied in large animals such as sheep [198,216,217], pigs [218], dogs [116], and non-human primates, including macaques [209,219]. Given the similarity to humans, studies on non-human primates are considered an important intermediate step in validating the safety and efficacy of ultrasound in human studies. Sheep have been an attractive model for ultrasound-mediated BBB opening, with the rationale that this species shares many important brain and skull features with humans [215,217]. The thickness, porosity, and curvature of the calvarium in sheep skulls are very close to that of humans [220,221]. This similarity also extends to the morphology of the brain; however, the sheep’s brain is smaller and its neuroanatomical structures are less homogeneous and gyrencephalic [222,223]. Sheep have a bodyweight profile closer to humans, and this is particularly important for optimizing the dosing and pharmacokinetics of drugs in relation to BBB disruption [198].

Ultrasound-induced BBB opening in small-to-large animal models has been used to study enhanced delivery of a variety of chemotherapeutic agents [54,202] such as herceptin [139], liposomal doxorubicin [52], cytarabine [224], doxorubicin [200], temozolomide [225], and methotrexate [226], as well as genes [227,228,229,230], viruses [228], and cells [207]. This concept, as well as relevant animal models, chemotherapeutic drugs, and the acoustic parameters used for BBB opening, have been recently reviewed [54,126,202]. Beyond BBB modification, ultrasound sonication has shown significant beneficial effects for several other neurological diseases, including seizure and epilepsy, in non-human primates [231,232,233,234].

### 1.8. Ultrasound Devices Developed for Clinical Application of BBB Opening with Low-Intensity Pulsed Ultrasound

Despite promising findings from pre-clinical animal studies and the considerable progress in clinical translation of ultrasound-induced BBB opening, selective and localized access to target areas in the human brain remains a major challenge. This is particularly due to the thickness of the skull and the heterogeneity of the human brain.

The human skull consists of two layers of cortical bone, separated by a central layer of cancellous bone (diploë) consistent of liquid-filled pores, which reflect and distort ultrasound waves, thus dampening the pressure wave and increasing the liberation of heat into the skull [235]. Furthermore, the sound velocity distortion and phase aberrations, as well as high absorption, can rapidly lead to poor focusing and higher energy loss, especially at higher ultrasound frequencies [236]. During the past two decades, the search for a way to obtain a uniform ultrasound beam through the skull has been ongoing [237]. In this context, HIFU is often applied, in which the ultrasound-induced BBB opening mainly depends on the thermal effects and the beam intensity of ultrasound. In contrast, FUS-induced BBB opening relies mostly on mechanical effects of ultrasound such as cavitation, which depends on the beam pressure and is thus inherently less concentrated than thermal effects [215].

Therapeutic FUS has recently been translated to the clinic with either extracranial non-invasive devices or minimally invasive implantable devices [238]. The principal attraction of these extracranial non-invasive techniques is their ability to exert biological effects through an intact skull [239]. However, the skull represents the main obstacle for the application of ultrasound, because bone strongly attenuates, reflects, and distorts ultrasound, resulting in inefficient delivery, off-target effects, and rapid heating within. Furthermore, the thickness and density of the skull widely vary in different locations and between individuals [240,241]. The presence of hair on the stimulation site can reduce ultrasound delivery up to 80% depending on several parameters including characteristics of the hair and the ultrasound frequency [242]. A variety of external and implantable ultrasound systems recently developed are currently in clinical trials (Figure 7).

The ExAblate system (Figure 7A) is a transcranial, magnetic resonance, and MRgFUS system developed by InSightec as a tool for less-invasive thermal ablation of brain tissues for the treatment of medication-refractory tremors in patients with essential tremor and tremor-dominant Parkinson’s disease [243]. Beyond its initial purpose, several clinical trials have and are currently investigating its application for BBB opening in adult patients with high-grade gliomas and breast cancer brain metastases [238]. Using MRI guidance, a series of standard diagnostic MR images are taken to identify the location and shape of the structure to be treated. The resulting images are loaded into the ExAblate workstation and utilized to graphically determine the region of interest. The ExAblate system uses this neuronavigational approach to focus ultrasound energy in targeted regions and then repeatedly transmit ultrasound until the desired outcome is achieved, confirmed via MRI immediately after the treatment. The MR thermometry feedback information during treatment is analysed by the physician to monitor patient safety and to control and adapt system parameters for optimal results.

NaviFUS (Figure 7B), designed by a Taiwanese biotech company, is another extracranial, multichannel hemispheric phased-array ultrasound system. This device has been recently assessed in a single-arm dose-escalation study in patients with recurrent GBM to bypass the BBB and improve drug delivery to the brain [244].

An extracranial single-element focused ultrasound system (Figure 7C; Sonic Concepts, WA, USA) has recently been approved by the FDA for a pilot clinical trial on cognitive function in Alzheimer’s disease (Columbia University, NCT04118764). The treatment procedure is directed by a neuronavigational system and a passive cavitation detection device.

**Figure 7 pharmaceutics-14-02231-f007:**
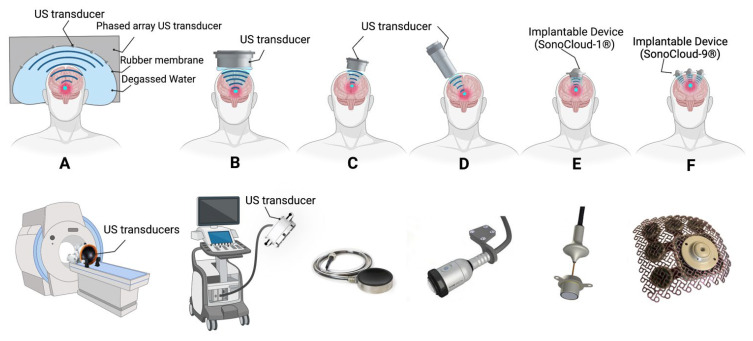
Schematic representation of ultrasound devices developed for clinical application of BBB opening and treatment of neurological conditions with low-intensity pulsed ultrasound. (**A**,**B**) Extracranial hemispheric focused ultrasound arrays (multi-element devices, ExAblate^®^ and NaviFUS^®^, respectively). (**C**) Extracranial mono-element focused device. (**D**) Transcranial Pulse Stimulator with real-time tracking system (TPS^®^, NEUROLITH). (**E**) Implantable, unfocused single-emitter ultrasound device (SonoCloud-1^®^). (**F**) Implantable, unfocused nine-emitter ultrasound device (SonoCloud-9^®^). Adapted with permission of Elsevier from ref [245], permission conveyed through Copyright Clearance Center, Inc.

Transcranial Pulse Stimulation (TPS^®^, NEUROLITH; Figure 7D) is a revolutionary new ultrasound device based on real-time monitoring which delivers short acoustic pulses with an ultrasound frequency range. TPS^®^, produced by STORZ Medical AG in Switzerland, has several advantages over the other currently available technologies, including no requirements for immobilization of the patient and shaving of the scalp. In addition, using the real-time tracking of the handpiece position allows the automatic visualization of the treated regions. Upon changing the handpiece position, the visualization of the target regions in the loaded MRI scans is automatically updated and the applied energy is also highlighted in colour. Owing to the short duration of the TPS^®^ stimulation, tissue heating is avoided. The key mechanism of focused stimulation of deep cerebral regions (up to 8 cm) by TPS^®^ is mechano-transduction. Several ongoing clinical trials are investigating the efficacy and safety of the TPS^®^ in adults with Parkinson’s and Alzheimer’s disease [239,246]. Nevertheless, this ultrasound device has not been used for BBB opening in clinical studies so far.

SonoCloud-1 (Figure 7E) and SonoCloud-9 (Figure 7F) are the two main implantable ultrasound devices for brain stimulation and BBB opening in clinical trials. Developed by CarThera, the SonoCloud-1 device [83,247] is an implantable unfocused ultrasound device which can be inserted in a burr hole and activated using a transcutaneous connection. The SONOKID clinical trial started in 2020 in France to evaluate the safety and feasibility of BBB opening in the paediatric population [248] to improve the efficacy of carboplatin chemotherapy in recurrent supratentorial malignant primitive tumours. So far, SONOKID is known as the only clinical trial on ultrasound-induced BBB opening in the paediatric population.

The SonoCloud-9 device [249], a recent upgrade on the SonoCloud-1 device, was designed to improve treatment coverage and volume. This system is designed for transcranial sonication of brain tumours and surrounding infiltrative regions in patients with GBM. The device has an implantable array of nine 1 cm (diameter) transducers on a grid which provides a higher treatment volume compared with SonoCloud-1. The ultrasound transducers are installed through a craniotomy in the skull at target sites, and connections are made to the transducers using needle electrodes passed through the scalp acutely. This system is currently being studied in an international multicenter clinical trial in patients with recurrent GBM. Recently, SonoCloud-9 received FDA approval for phase I and II clinical trials in the United States [250].

Each system has its own advantages and disadvantages, and these techniques can be complementary depending on the particular indication and anatomical location of interest. Extracranial devices are non-invasive, deep-penetrating, and safe [251]. They require 100 times lower acoustic power values than those needed to produce thermal damage in tissue [204]. However, performing the sonication process is much longer (2 to 4 h) and covers a limited volume of the brain (1 to 4 cm^3^). In addition, extracranial devices are accompanied by theorized difficulties in focusing on the superficial targets in the brain. On the other hand, implantable devices are fast (4 to 15 min) and larger volumes (4 to140 cm^3^) of BBB opening can be achieved. However, application of these systems, requires surgical implantation via a burr hole during a tumour debulking or biopsy procedure, and the targeted volume remains fixed and size dependent on the device.

Therefore, the clinical applications of these systems may vary. Large, superficial, and infiltrative lesions such as extensive high-grade glioma could be a good implication for the implantable devices, while smaller and deeper lesions such as hypothalamic or basal ganglia lesions can be treated with extracranial devices [238].

### 1.9. Ultrasound-Mediated Therapies in Clinical Trials

Pioneering research has shown that FUS can temporarily and repeatedly disrupt the BBB in a targeted fashion without open surgery in humans. Currently, the majority of FUS-mediated BBB opening trials being conducted in humans are aimed at delivering chemotherapeutic agents to treat brain tumours (Table 1) [211].

Thermal ablation (the process of removing tissue using the heat generated by HIFU) by transcranial non-invasive FUS is potentially the most straightforward approach for brain tumour therapy. However, achieving adequate tumour necrosis and minimal off-target effects remains the major challenge [243,252]. Low-intensity ultrasound delivers only <0.1% of the energy that is needed to result in thermal ablation. Such ultrasound irradiation can be utilized for less-invasive and temporary BBB opening, particularly when it is combined with MBs [253]. Due to the lower energy required, the targeted area can be expanded, and adjusted based on the target shape and location within the intracranial vault [254].

Two clinical trial studies (study no. NCT02253212, NCT03744026) have been performed in adults with recurrent GBM treated with intravenous carboplatin, using the pulsed ultrasound systems SonoCloud-1 or SonoCloud-9, respectively. These trials have indicated that repeated transient BBB opening, in combination with systemic MB injection, is safe and has the potential to optimize chemotherapy delivery such as carboplatin in the brain [83,247]. The BBB opening procedures were well-tolerated, without severe adverse events, including during the sonication of specific brain regions. Both median progression-free survival (PFS) and overall survival (OS) of the patients were significantly improved when compared to the historical data (4.11 months vs. 2 to 3 months for PFS and 12.94 months vs. 6 to 9 months for OS, respectively). An ongoing clinical trial in France is investigating the safety and efficacy of BBB opening using the SonoCloud-1 in 21 patients with melanoma brain metastases (NCT04021420).

Another clinical trial of low-intensity FUS-BBB opening with systemically administered liposomal doxorubicin or temozolomide in patients with malignant brain tumours has been reported (NCT02343991) [254]. The BBB within the target volume showed radiographic evidence of opening with an immediate 15 to 50% increased contrast enhancement on MRI and the concentrations of temozolomide and, to a lesser extent, doxorubicin increased in sonicated tissue. BBB disruption was transient, and the integrity of the BBB was shown to be restored approximately 24 h after sonication. The procedure was well-tolerated with no adverse clinical symptoms related to the procedure. In addition, several ongoing clinical trials are assessing the safety and feasibility of low intensity FUS-mediated BBB opening when combined with chemotherapies for primary brain tumours and breast cancer brain metastases (Table 1).

As mentioned above, another application of ultrasound-mediated BBB opening could be for treating CNS neurological disorders treatment such as amyotrophic lateral sclerosis (ALS), Parkinson’s and Alzheimer’s disease (summarized in Table 1). Interestingly, even without the administration of therapeutic agents, ultrasound-mediated BBB opening has shown significant beneficiary effects in several pre-clinical and clinical Alzheimer’s studies [255,256,257].

A recent Phase I safety and feasibility trial (NCT03739905) in five patients with early-to-moderate Alzheimer’s disease demonstrated for the first time that the BBB within the target volume was safely, reversibly, and repeatedly opened. No clinically severe adverse events nor clinically significant worsening in cognitive performance 3 months after opening the BBB were observed [10]. Ultrasound is also actively being investigated to deliver treatments in other diseases, such as ALS (NCT03321487) [254] and Parkinson’s disease (NCT03608553, NCT04370665) [258,259]. One of the limitations of current clinical trials is the lack of information on the efficacy of BBB opening on brain drug delivery. In addition, most of these studies are often uncontrolled and have a limited number of participants, and are predominantly Phase I/II, thereby larger controlled trials are required to evaluate the safety and efficacy of BBB opening.

## 2. Conclusions

Despite the meteoric developments in brain tumour drug delivery, overcoming BBB and BTB is a major challenge which limits the efficacy of these treatments. In addition, most approaches result in new local and systemic side effects. Among all of these approaches, ultrasound-mediated drug delivery holds great potential for designing non-invasive, targeted pharmacologic neuro-interventions. However, despite the promising pre-clinical and clinical findings, there are several limitations with currently available ultrasound-mediated drug delivery systems. Perhaps the most pressing of these includes the need to develop effective devices for precise targeted local delivery of ultrasound, which allow adequate doses of ultrasound to reach the targeted site. In addition, the development of intelligent sono-responsive vehicles for drug delivery is another key factor for ultrasound-mediated drug delivery to succeed. The development of smart carriers that can sequester cytotoxic anti-cancer drugs, lower their systemic side effects and hold onto the payload drug unless release is triggered by ultrasound is urgently required. Nevertheless, the use of low-power ultrasound in therapeutic medicine is in its infancy and it is expected to see a continued increase in the development of more sophisticated ultrasound-responsive drug delivery systems for individualizing chemotherapy for brain tumours and other solid human tumours.

## Figures and Tables

**Figure 1 pharmaceutics-14-02231-f001:**
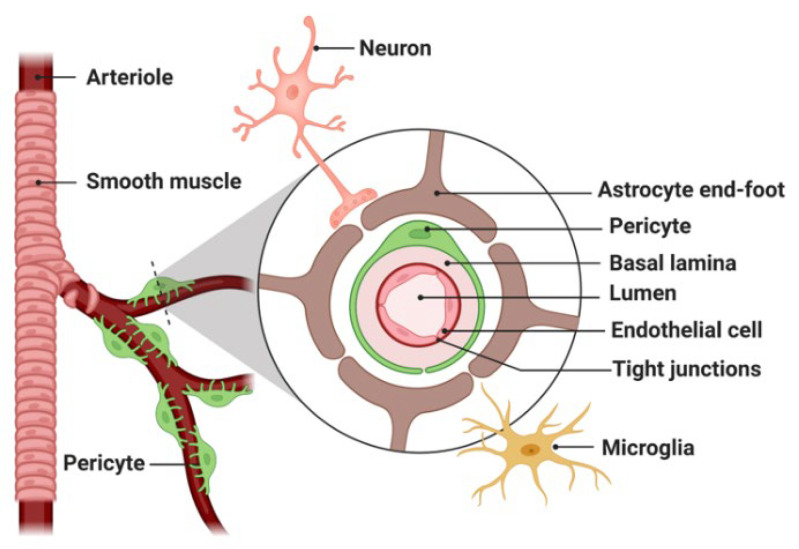
Schematic representation of blood–brain barrier (BBB) composition. The BBB comprises vascular endothelial cells sealed by tight junctions, covered by pericytes embedded in the basal lamina, astrocyte end-feet, perivascular microglia/macrophages, and neurons. (Created with BioRender.com).

**Figure 2 pharmaceutics-14-02231-f002:**
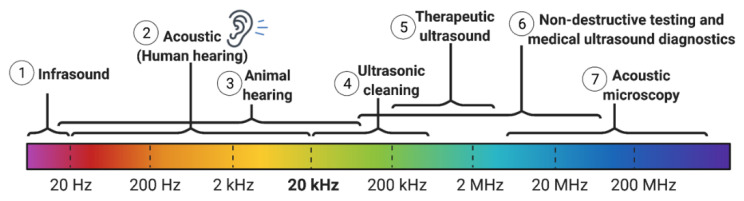
The spectrum of ultrasound with respect to audible sound and different frequency ranges for diagnostic versus therapeutic applications. (Created with BioRender.com).

**Figure 6 pharmaceutics-14-02231-f006:**
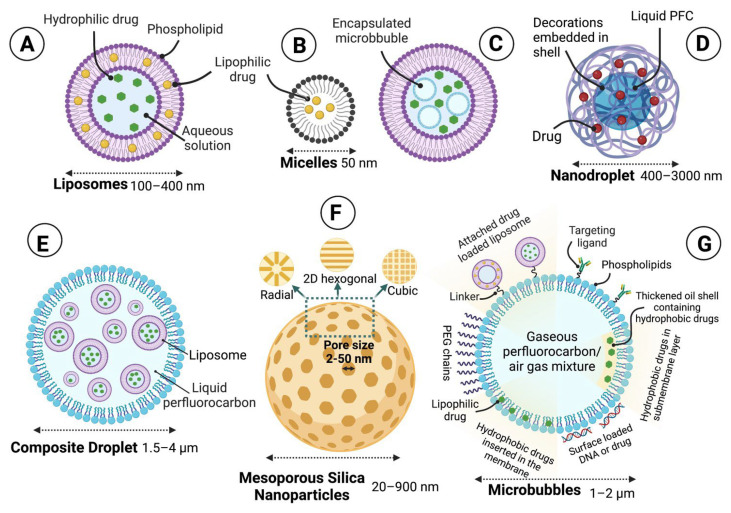
Schematic representation of various ultrasound-sensitive carriers. Liposomes (**A**) can deliver hydrophobic or hydrophilic drugs after ultrasound exposure. Micelles (**B**) can carry hydrophobic drugs within their core and have a higher release profile in response to ultrasound. (**C**) The MB can be encapsulated within a liposome along with the drug. When exposed to ultrasound, the MB ruptures the outer liposome, releasing the payload. Nanodroplets (**D**) can absorb acoustic energy, which causes the liquid perfluorobutane core to phase-change into a gas microbubble, thereby leading to drug release through the development of mechanical forces to the cellular membrane. Composite-droplets (**E**) are multiple emulsions (water or oil-in perfluorocarbon-in water) that can be converted with an imaging scanner and can transport large payloads. Mesoporous silica nanoparticles (**F**) are silica nanoparticles with variable pore structures such as radial, hexagonal, and cubic for controlled release. The grafted ultrasound-responsive polymers on the surface, acting like a nano valve, can control the release of loaded therapeutic molecules from these nanoparticles. Finally, MBs (**G**) can be covalently linked to drug-filled liposomes, or nucleic acids can be adsorbed on the external surface of the MBs via electrostatic attractions. The membrane can be thickened with an oil layer, allowing hydrophobic drugs to be carried within it or incorporated into the lipid monolayer shell of the MB. Additionally, targeting ligands such as antibodies can be conjugated to the surface to help facilitate the accumulation of MBs in desired tissues (Created with BioRender.com).

**Table 1 pharmaceutics-14-02231-t001:** A summary of clinical trials to date evaluating the effects of ultrasound-mediated blood–brain barrier opening in humans (ClinicalTrials.gov, 2021). GBM, Glioblastoma multiforme; MRgFUS, MR-guided Focused; US, ultrasound. Permission for adaptation from ref [211] granted under Creative Commons Attribution 4.0 International License, (http://creativecommons.org/licenses/by/4.0/; accessed on 28 June 2022).

	Trial Number	Study Title, Date	Condition	Interventions	Number of Participants	Therapeutic Protocol	Location	Status
Brain tumours								
**1**	NCT02253212	Safety of BBB Opening With the SonoCloud (SONOCLOUD), 2014	Glioma or GBM	Device: SonoCloud^®^ Drug: Carboplatin and SonoVue^®^ microbubble	27	The US (0.5–1.1 Mpa) was activated monthly before IV administration of carboplatin and microbbuble (0.1 mL/kg) (min. 6 cycles) [83].	France	Completed
**2**	NCT02343991	Blood–Brain Barrier Disruption Using Transcranial MRI-Guided Focused US, 2014	Primary brain tumours	Device: ExAblate MRgFUS Drug: Liposomal doxorubicin or Temozolomide with Definity^®^ microbubbles	10	Five patients underwent the MRgFUS in conjunction with administration of chemotherapy (*n* = 1 liposomal doxorubicin, *n* = 4 temozolomide) one day prior to surgical resection. Samples of “sonicated” and “unsonicated” tissue were collected during surgery [242].	Canada	Active, not recruiting
**3**	NCT03712293	ExAblate Blood–Brain Barrier Disruption for Glioblastoma in Patients Undergoing Standard Chemotherapy, 2018	GBM	Device: Transcranial ExAblate 4000 Type 2.0 MRgFUS Drug: Temozolomide	10	The ExAblate BBB disruption will coincide with one of three first days of each planned temozolomide adjuvant therapy cycle as one procedure per cycle.	South Korea	Recruiting
**4**	NCT03626896	Safety of BBB Disruption Using NaviFUS System in Recurrent Glioblastoma Multiforme (GBM) Patients, 2018	Glioma or GBM	Device: NaviFUS^®^ System Drug: None.	6	The study will be carried out in patients with recurrent GBM who will undergo surgery within 2 weeks to evaluate the safety and the tolerated US dose (escalated exposure average 10–16 W).	Taiwan	Completed
**5**	NCT03616860	Assessment of Safety and Feasibility of ExAblate Blood–Brain Barrier (BBB) Disruption for Treatment of Glioma, 2018	GBM	Device: ExAblate 4000 Type 2.0 MRgFUS Drug: Temozolomide and Definity^®^ microbubbles	20	Patients will undergo up to 6 treatments with FUS coincident with their standard temozolomide cycles.	Canada	Recruiting
**6**	NCT03714243	Blood–Brain Barrier Disruption (BBBD) Using MRgFUS in the Treatment of Her2-positive Breast Cancer Brain Metastases (BBBD), 2019	Metastatic HER-2 positive breast cancer	Device: ExAblate 4000 Type 2.0 MRgFUS Drug: Trastuzumab	10	Six study ExAblate BBB opening treatment cycles, every 2–3 weeks based on their trastuzumab regimen.	Canada	Recruiting
**7**	NCT03744026	Safety and Efficacy of Transient Opening of the Blood–Brain Barrier (BBB) With the SonoCloud-9 (SC9-GBM-01), 2019	GBM	Device: SonoCloud-9 Drug: Carboplatin	33	Patients will undergo 6 cycles of carboplatin treatments every 4 weeks coincident with BBB opening in resection area and surrounding tissues using SonoCloud-9 system [238].	France	Recruiting
**8**	NCT03551249	Assessment of Safety and Feasibility of ExAblate Blood–Brain Barrier (BBB) Disruption, 2019	Glioma or GBM	Device: ExAblate 4000 Type 2.0 MRgFUS Drug: Temozolomide	20	BBB will be disturbed along the periphery of tumour resection cavity prior to beginning the planned adjuvant temozolomide chemotherapy phase of treatment.	USA	Recruiting
**9**	NCT04021420	Safety and Efficacy of Sonocloud Device Combined With Nivolumab in Brain Metastases From Patients With Melanoma (SONIMEL01), 2019	Metastatic melanoma	Device: SonoCloud^®^ Drug: Nivolumab Injection alone or with Ipilimumab	21	Along with systemic injection of an US resonator and prior to beginning the chemo-treatment, SonoCloud^®^ delivers US for a duration of 120–270 s. A total of 3 US dose levels will be evaluated (0.78, 0.9 and 1.03 MPa).	France	Recruiting
**10**	NCT04528680	US-based Blood–Brain Barrier Opening and Albumin-bound Paclitaxel for Recurrent Glioblastoma (SC9/ABX), 2020	GBM	Device: SonoCloud-9 Drug: Albumin-bound paclitaxel (Abraxane^®^), microbubbles	39	The device will be implanted at the time of surgical resection of the recurrent tumour. During that procedure, a first test dose of the chemotherapy will be administered in the operating room after sonication and tissue concentrations in different parts of the resected tumour will be measured. In select patients, the sonication procedure will occur immediately after the test dose of chemotherapy is administered.	USA	Recruiting
**11**	NCT04614493	Innovative SonoCloud-9 Device for Blood–Brain Barrier Opening in First Line Temozolomide Glioblastoma Patients. (SonoFIRST), 2020	GBM	Device: SonoCloud-9 Drug: Temozolomide	66	The patients will receive daily temozolomide during Radiation, followed by 6 months of adjuvant temozolomide 5 days/months) with 6 concomitant BBB opening sessions by US + 9 BBB opening sessions by US without any associated drug.	Belgium/France	Not yet recruiting
**12**	NCT04440358	ExAblate Blood–Brain Barrier Disruption With Carboplatin for the Treatment of rGBM, 2020	GBM	Device: ExAblate 4000 Type 2.0 MRgFUS Drug: Carboplatin with microbubble	50	Patients will undergo up to 6 cycles of ExAblate BBBD procedures in conjunction with carboplatin chemotherapy about every 4 weeks.	South Korea	Recruiting
**13**	NCT04417088	ExAblate Blood–Brain Barrier Disruption for the Treatment of rGBM in Subjects Undergoing Carboplatin Monotherapy, 2020	GBM	Device: ExAblate 4000 Type 2.0 MRgFUS Drug: Carboplatin with microbubble	30	Patients will undergo up to 6 cycles of ExAblate BBBD procedures in conjunction with carboplatin chemotherapy about every 4 weeks.	USA	Recruiting
**14**	NCT04804709	Non-Invasive Focused US (FUS) With Oral Panobinostat in Children With Progressive Diffuse Midline Glioma (DMG), 2021	Diffuse Midline Glioma	Device: FUS treatment with neuro-navigator-controlled sonication Drug: Panobinostat, microbubbles	15	After each instance of opening the BBB using specific parameters of FUS in the specific number of tumour sites (one, two, or three), the subjects will receive oral Panobinostat.	USA	Recruiting
**15**	NCT04063514	The Use of Focused US and DCE K-trans Imaging to Evaluate Permeability of the Blood–Brain Barrier, 2025	Glioma	Device: Brainsonix FUS and DWL Doppler system Drug: Definity^®^ microbubbles	15	Not mentioned.	USA	Not yet recruiting
**Alzheimer’s Disease**								
**1**	NCT02986932	Blood–Brain Barrier Opening Using Focused US With IV Contrast Agents in Patients With Early Alzheimer’s Disease (BBB-Alzheimers), 2016	Alzheimer’s Disease	Device: ExAblate MRgFUS Drug: Definity^®^ microbubbles	6	In the first stage, patients will undergo a small area BBB opening (9 × 9 mm) with multiple sonications to establish the minimum required sonication parameters. In stage II, a larger volume (2.5–3.0 cm) will be targeted.	Canada	Completed
**2**	NCT03119961	Blood–Brain Barrier Opening in Alzheimer’s Disease (BOREAL1), 2017	Alzheimer’s Disease	Device: SonoCloud^®^, CarThéra Drug: anti- Alzheimer’s Disease drugs	10	Not mentioned.	France	Completed
**3**	NCT03671889	ExAblate Blood–Brain Barrier (BBB) Disruption for the Treatment of Alzheimer’s Disease, 2018	Alzheimer’s Disease	Device: ExAblate 4000 Type 2.0 MRgFUS Drug: Not mentioned	20	Three serial ExAblate BBB disruption procedures in specific areas in the brain will be carried out.	USA	Recruiting
**4**	NCT03739905	ExAblate Blood–Brain Barrier Opening for Treatment of Alzheimer’s Disease, 2018	Alzheimer’s Disease	Device: ExAblate 4000 Type 2.0 MRgFUS Drug: Alzheimer’s medication	30	Three serial ExAblate BBB disruption procedures in specific areas in the brain will be carried out.	Canada	Recruiting
**5**	NCT04118764	Non-invasive Blood–Brain Barrier Opening in Alzheimer’s Disease Patients Using Focused US, 2020	Alzheimer’s Disease	Device: Neuronavigation-guided single-element focused US transducer Drug: Definity^®^ microbubbles	6	Patients will undergo a FUS treatment to the brain, along with Magnetic Resonance Imagine [128] with or without gadolinium contrast agents and Positron Emission Tomography (PET) scans.	USA	Recruiting
**6**	NCT04526262	Assessment of Initial Efficacy and Safety of High Intensity Focused US ‘ExAblate 4000 Type 2′ for Blood–Brain Barrier Disruption in Patients With Alzheimer’s Disease, 2020	Alzheimer’s Disease	Device: ExAblate 4000 Type 2.0 MRgFUS Drug: Alzheimer’s medication	6	Two sessions of transcranial magnetic resonance-guided focused US blood–brain barrier disruption every 3 months.	South Korea	Active, not recruiting
**Other**								
**1**	NCT03608553	A Study to Evaluate Temporary Blood–Brain Barrier Disruption in Patients With Parkinson’s Disease Dementia, 2018	Parkinson’s Disease with dementia	Device: ExAblate 4000 Type 2.0 MRgFUS Drug: Luminity^®^	10	In the first stage, patients will undergo a small area BBB opening (9 × 9 mm) with multiple sonications to establish the minimum required sonication parameters. In stage II, a larger volume (2.5–3.0 cm) will be targeted.	Spain	Active, not recruiting
**2**	NCT03321487	Blood–Brain Barrier Opening Using MR-Guided Focused US in Patients With Amyotrophic Lateral Sclerosis, 2018	Amyotrophic Lateral Sclerosis (ALS)	Device: ExAblate MRgFUS Drug: None	8	BBB will be disturbed using US in conjunction with an intravenous US contrast agent.	Canada	Active, not recruiting
**3**	NCT04370665	Blood–Brain Barrier Disruption With Cerezyme in Patients With Parkinson’s Disease, 2020	Parkinson’s Disease	Device: ExAblate MRgFUS Drug: Cerezyme^®^ (an analogue of the human enzyme beta-glucocerebrosidase)	4	Patients will undergo three biweekly delivery of Cerezyme^®^ via MRgFUS induced BBB opening to unilateral putamen.	Canada	Active, not recruiting

## Data Availability

All data supporting the reported conclusions can be found in the manuscript.

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
