# Peer review of "Harnessing Ultrasound for Targeting Drug Delivery to the Brain and Breaching the Blood–Brain Tumour Barrier"

_pharmaceutics, 2022, doi:10.3390/pharmaceutics14102231_

Round 1

Reviewer 1 Report

This review describes the recent achievements in ultrasound-targeted therapy to the brain. In my opinion the work is well written and presents in sophisticated and clear manner novel therapies ofbrain tumours. The Authors provided valuable figures and tables. There are also proper and actual references. The manuscript can be accepted in the present form.

Author Response

Comments and Suggestions for Authors: This review describes the recent achievements in ultrasound-targeted therapy to the brain. In my opinion the work is well written and presents in sophisticated and clear manner novel therapies of brain tumours. The Authors provided valuable figures and tables. There are also proper and actual references. The manuscript can be accepted in the present form.

Response:  Thank you for your time and consideration.

Reviewer 2 Report

The authors review literature describing the utility of ultrasound as an approach to enhance drug delivery to the brain. Overall, the review is well organized and comprehensive. I have the below comments for the authors.

1.     The below mentioned has been stated in the introduction section (line 31).

“Although it is well-established that vascular permeability in tumour tissue (known as the blood–tumour barrier; BTB) is greater than the BBB, the BTB compromises much of the structural characteristics and integrity of the BBB, contributing to suboptimal drug accumulation in brain tumours [4-7].”

The statement is unclear and should be rephrased for better clarity. Of late, it is fairly appreciated in the field that suboptimal drug exposures in regions of brain tumors is a critical challenge, although the BTB is compromised to some degree in certain regions of tumors in the brain. While the intent here is possibly to state this, it is unclear as written.

2.     Suggest replacing “leaky vasculature of BTB” (line 206) with something along the lines of ‘relatively leaky vasculature in some regions of the tumor’. While the BTB can be more permeable in brain tumors, a high degree of heterogeneity exists across regions within the same tumor.

3.     Section 1.6 discusses the historical perspective of ultrasound applications in a very elaborate manner. The authors should consider shortening this section and include a succint discussion.

4.     Line 453 states “down-regulation of P-gp drug efflux pumps expression along the endothelial cells membrane”. A similar statement in made in Line 821 as well.

The effect of focused ultrasound and microbubbles on BBB transporters and the mechanisms involved are not well understood. The authors should clearly state this for better clarity. 

While P-gp has been assessed in preclinical studies, further studies on focused ultrasound and its effects on different BBB transporters such as BCRP are needed for a more complete understanding. Moreover, preclinical literature examples suggest lack of significant improvement in brain drug exposures for small molecule inhibitors such as erlotinib that are substrates of active efflux.

5.     Authors should clearly specify if the stated results are from in vitro or in vivo studies as well as human or preclinical models throughout the review. For instance, it is unclear in line 501 if the statement is referring to in vivo studies.

6.     Section 1.7.3 seems to be rather elaborate and somewhat detracts from the main topic of the review. The authors should consider replacing with a more concise section.

7.     Recommend authors to re-evaluate the utility of Figures 7 and 8, and consider removing them. While the other figures seem to be useful, Figures 7 and 8 seem to be of little value. Also figure 6 and 8 are somewhat redundant. Alternatively, Figures 6, 7 and 8 could be combined in to a single figure.

8.  The authors touch upon drug delivery of large molecules but the review is heavily focused on small molecule therapeutics. While that may be adequate, including some more discussion on large molecule drug delivery would further improve the quality of the manuscript.

9.  Relevant references need to be included to support the statements made throughout the review. For instance, section 1.5 has detailed discussion on “Physics of Ultrasound” but does not cite any references.

10.  There seem to be some typographical errors that need correction. Following are a couple of examples: (a)  Energy misspelt in Figure 3, (b) statement in  Line 232.

Author Response

Reviewer 2

Comments and Suggestions for Authors: The authors review literature describing the utility of ultrasound as an approach to enhance drug delivery to the brain. Overall, the review is well organized and comprehensive.  I have the below comments for the authors.

Response: Thank you for your attention to our manuscript.

Comment 1. The below mentioned has been stated in the introduction section (line 31).

“Although it is well-established that vascular permeability in tumour tissue (known as the blood–tumour barrier; BTB) is greater than the BBB, the BTB compromises much of the structural characteristics and integrity of the BBB, contributing to suboptimal drug accumulation in brain tumours [4-7].”

The statement is unclear and should be rephrased for better clarity. Of late, it is fairly appreciated in the field that suboptimal drug exposures in regions of brain tumors is a critical challenge, although the BTB is compromised to some degree in certain regions of tumors in the brain. While the intent here is possibly to state this, it is unclear as written.

Response:  Thank you, the paragraph is rephrased as below.

[Line 31] “Vascular permeability within a brain tumour (known as the blood–tumour barrier; BTB) is heterogeneous and is on average greater than the BBB. This results in penetration of the anti-cancer drug through the BTB and into tumour tissue. However, the increase in drug permeability is not enough to achieve an effective tumour therapy [4-7].”

Comment 2. Suggest replacing “leaky vasculature of BTB” (line 206) with something along the lines of ‘relatively leaky vasculature in some regions of the tumor’. While the BTB can be more permeable in brain tumors, a high degree of heterogeneity exists across regions within the same tumor.

Response: Thank you for your suggestion. Changed as recommended, line 216.

Comment 3. Section 1.6 discusses the historical perspective of ultrasound applications in a very elaborate manner. The authors should consider shortening this section and include a succinct discussion.

Response: Thank you for your suggestion. Reduced in length as recommended.

Comment 4. Line 453 states “down-regulation of P-gp drug efflux pumps expression along the endothelial cells membrane”.  A similar statement in made in Line 821 as well.

The effect of focused ultrasound and microbubbles on BBB transporters and the mechanisms involved are not well understood. The authors should clearly state this for better clarity.

While P-gp has been assessed in preclinical studies, further studies on focused ultrasound and its effects on different BBB transporters such as BCRP are needed for a more complete understanding. Moreover, preclinical literature examples suggest lack of significant improvement in brain drug exposures for small molecule inhibitors such as Erlotinib that are substrates of active efflux.

Response: We agree, the statement at line 821 has been deleted. As requested, to clarify the paragraph,­­­­ we added:

[Line 459] “While P-gp has been assessed in preclinical studies, further studies on focused ultrasound and its effects on different BBB transporters such as BCRP are needed for a more complete understanding of such mechanisms” to the document.

Comment 5. Authors should clearly specify if the stated results are from in vitro or in  vivo studies as well as human or preclinical models throughout the review. For instance, it is unclear in line 501 if the statement is referring to in vivo studies.

Response: The manuscript has been adjusted accordingly.

[Line 420] “In 1995, Vykhodtseva et al. reported that ultrasound can reversibly improve BBB permeability in rabbit brain in vivo without significant damage of the BBB or brain parenchyma [112].”

[Line 511] “Synergistic cytotoxic effects were observed in many in vivo and in vitro studies when ultrasound was added to the chemotherapeutic drugs (reviewed by Pitt. WG. et al. [125]).”

[Line 643] “In addition, liposomal chemotherapeutic formulations have reduced systemic cytotoxicity when compared to free drug, an example being liposomal doxorubicin which has reduced cardiotoxic effects in clinical studies [178].”

Comment 6.  Section 1.7.3 seems  to  be  rather  elaborate  and  somewhat  detracts  from  the  main  topic  of  the review. The authors should consider replacing with a more concise section.

Response: Thank you for the comment. Our review aims to cover the area of ultrasound-responsive drug delivery. We think the story would not complete unless we have this section. In addition, Reviewer 3 of the manuscript requested us to include more details about some of the NPs. Nevertheless, we have removed the original Fig 6 and Fig 8. We hope the current version is more focused and concise than before.

Comment 7. Recommend authors to re-evaluate the utility of Figures 7 and 8 and consider removing them. While the other figures seem to be useful, Figures 7 and 8 seem to be of little value. Also figure 6 and 8 are somewhat redundant. Alternatively, Figures 6, 7 and 8 could be combined into a single figure.

Response: Thank you. We agree and removed Fig 6 and 8 from the section.

Comment 8. The authors touch upon drug delivery of large molecules but the review is heavily focused on small molecule therapeutics. While that may be adequate, including some more discussion on large molecule drug delivery would further improve the quality of the manuscript.

Response: Thank you for the comment. In this review, our focus is on sono-responsive drug delivery of therapeutic compounds for brain tumors and neurological conditions. We have avoided focusing on specific types of payload by size and how this influences their delivery by ultrasound since we feel that this topic is beyond the scope of the review and warrants separate consideration.

Comment 9. Relevant references need to be included to support the statements made throughout the review. For instance, section 1.5 has detailed discussion on “Physics of Ultrasound” but does not cite any references.

Response: Thank you, done. Specifically for section 1.5:

[Line 233] “Ultrasound is composed of sound-waves that exceed the audible range of human frequencies (>20 kHz, Figure 2), and functions as a pressure wave through a physical medium (e.g. air or water). Ultrasound waves are usually generated via a piezoelectric transducer, which converts electrical energy into mechanical movement [60]. Such waves can be absorbed, reflected, and/or bent, and unlike light waves require a physical medium for transmission. In other words, ultrasound waves constitute the oscillatory movement of molecules (about fixed points) in the medium at high and low pressure, corresponding to compression and rarefaction, respectively [60,61]. It is these mechanical mechanisms through which ultrasound is theorized to affect biomolecules and disrupt biological barriers. In comparison with light, ultrasound has significantly lower absorption by water and most soft-tissues, thus allowing it to penetrate deeply into the body and transmit energy to a precise location within target tissue [62]. Such properties of ultrasound are used for the design of diagnostic ultrasound scanners and therapeutic devices that currently are currently in clinical use.”

Comment 10. There seem to be some typographical errors that need correction.  Following is a couple of examples: (a) Energy misspelt in Figure 3, (b) statement in Line 232.

Response:  Thank you for noticing these. We agree and have corrected these errors (Figure 3 and line 244). The manuscript has been revised to minimise such errors.

Reviewer 3 Report

The paper by Barzegar-Fallah et al reviews the studies using focused ultrasound for drug delivery to the brain. This is a well written paper. However, my major concern is that the authors need to specify how their review is different than several other existing review papers in this field. What areas are they covering that have not been covered before in the prior reviews. This should be mentioned explicitly in the introduction section. After addressing the comments, I recommend publication of this paper.

1.       In the introduction, more focus has been given to liposomes. Other nanoparticles have been studied for drug delivery to the brain including nanobubbles and phase-shift nanodroplets. These also need to be mentioned.

2.       Line 125: what range of gaps between endothelial cells are reported in leaky vasculature? This will help understand the importance of size of practices used for tumor drug delivery.

3.       Lines 140, 141: Can the authors comment why they think nanobubbles and phase-shift nanodroplets are not among the common approaches.

4.       Line 199: the depth of penetration depends on the frequency.

5.       Line 201: what range of ultrasound parameters will generate hyperthermia. I think this will be a helpful piece of information to include for the three different complementary effects of ultrasound.

6.       Line 229: This is not correct. Ultrasound gets absorbed by the tissue depending on the acoustic parameters and tissue properties. If ultrasound is not absorbed how the thermal effect is justified? In section 1.6.1 the authors are stating that the acoustic energy is absorbed.

7.       Line 336: what do the authors mean by beyond diffusion? This is not clear to me?

8.       It would be helpful for the readers if authors report the corresponding acoustic pressure (i.e., peak negative and peak positive) wherever they mention the intensity. E.g. line 360 what does 100 W/cm^2 correspond to in terms of peak rarefactional and peak positive.

9.       Line 386: what frequency?

10.   Line 438: Does inertia cavitation cause transient or permanent permeability?

11.   Line 439: Please explain how theoretically it is expected that stable oscillation of bubbles cause cell damage? Are you referring to the generated shear stresses? Then mention the range.

12.   Line 537: Phase-shift nano and micro droplets, which were introduced in 2000, have shown much enhanced stability and payload retention before reaching to the target. There are several papers on using phase-shift droplets for drug delivery. I think discussing the advantages of nanodroplets to liposomes and nanobubbles in terms of both stability and acoustic responsiveness would improve this review further.

13.   Line 580: why is using an unfocused stimulation beneficial?

14.   Line 625: What range of sizes are liposome and micelles are typically fabricated and studied? I did not find this information throughout the paper.

15.   Line 710: Can the authors explain how the formulation of lipid bilayer can trigger either thermal or mechanical effects of ultrasound?

Author Response

Reviewer 3

Comments and Suggestions for Authors: The paper by Barzegar-Fallah et al reviews the studies using focused ultrasound for drug delivery to the brain. This is a well written paper.  However,  my major concern  is  that  the  authors  need  to specify  how  their  review  is  different  than  several  other  existing  review  papers  in  this  field.  What areas are they covering that have not been covered before in the prior reviews. This should be mentioned explicitly in the introduction section. After addressing the comments, I recommend publication of this paper.

Response: Thank you for your attention to our manuscript. We believe our manuscript covers a more comprehensive and up-to date review of the literature on the therapeutic applications of ultrasound. Our review covers from the basics of ultrasound and controlled drug delivery using ultrasound to preclinical models and clinical studies that used ultrasound as tool to modify the BBB. We also tried to elaborate the mechanisms by which ultrasound boosts the release of a payload from NPs, such as liposomes. We summarized most of the clinical trials in which the transducers were applied, in a table with a short description of the study, type of the instrument, drugs and protocols of the study. Although some of the topics might have been covered by other reviews, we aimed to complement these topics with more up to date details and mechanisms. For example, no previous review has included all the currently available ultrasound transducers. Thus our review is novel as it covers a wide range of areas, but with enough details to interest most readers.

[Line 63] “Here, we will provide a comprehensive review of the application of ultrasound in controlled drug delivery, from the basics of ultrasound and sono-responsive carriers to the clinical application of sophisticated ultrasound transducers for treating brain tumours. We will initially review the challenges in treating brain tumours, and strategies used to bypass the BBB/BTB to enhance drug delivery. Sono-responsive drug delivery, and therapeutic ultrasound devices developed for this purpose will then be reviewed in detail. Finally, pre-clinical models and clinical studies using ultrasound-mediated BBB opening for the treatment of brain cancers will be discussed.”

Comment 1. In the introduction, more focus has been given to liposomes. Other nanoparticles have been studied for drug delivery to the brain including nanobubbles and phase-shift nanodroplets. These also need to be mentioned.

Response: Thank you, we agree and have modified the manuscript accordingly. We reduced the emphasis on liposomes and included phase-shift nanodroplets through the document. Also, microbubbles were broadened to nano/microbubble.

[Line 741] “In recent years, the emerging phase-changing ultrasound-responsive nanodroplets have attracted substantial attention for imaging and tumour therapy. Ultrasound-responsive phase change in the structure of NPs occurs through a process known as acoustic droplet vaporization (ADV). Generally, these NPs are made of a PFC core and a shell coating. The acoustic excitation of such NPs results in vaporization of the core into gas bubbles and a build up a mechanical pressure for sonoporation of cell membranes [195].

Comment 2. Line 125: what range of gaps between endothelial cells are reported in leaky vasculature? This will help understand the importance of size of practices used for tumor drug delivery.

Response: Thank you. We have not been able to find reported quantitative values of gap sizes between adjacent endothelial cells in the BTB, as compared to the BBB. Instead we have included reported data on increased transport of drug molecules, and their respective molecular weights, across the BTB compared to the BBB.

 [Line 129] “For perspective, increased transport of normally impermeably drug molecules such as liposomal doxorubicin (580 Da) [28] and datasinab (488 Da) [29] have been reported across the BTB as compared to regions with an intact BBB.”

Comment 3. Lines  140,  141:  Can  the  authors  comment  why  they  think  nanobubbles  and  phase-shift nanodroplets are not among the common approaches.

Response: Thank you for your question. We agree and acknowledge our omission. In the revised version, we have added nanobubbles and phase-shift nanodroplets to the sentence (Line 149) and later (Line 741) and modified figure 6 also.

[Line 741] “In recent years, the emerging phase-changing ultrasound-responsive nanodroplets have attracted substantial attention for imaging and tumour therapy. Ultrasound-responsive phase change in the structure of the NPs occurs through a process known as acoustic droplet vaporization (ADV). Generally, these NPs are made of a PFC core and a shell coating. The acoustic excitation of such NPs resulting in vaporization of the core into gas bubbles and a build up of mechanical pressure for sonoporation of cell membranes [195]. Phase shift droplets offer many advantages over micro/nano-bubbles and liposomes due to their higher stability and possibility of smaller sizes [196]. Micro/nano-bubbles and liposomes also suffer from limitations such as low spatial selectivity and short circulation time in vivo [195,197].”

Figure 6. “Schematic representation of various ultrasound-sensitive carriers. … Nanodroplets (D) can absorb acoustic energy which causes the liquid perfluorobutane core to phase-change into a gas microbubble, thereby leading to drug release through the development of mechanical forces to the cellular membrane. ….”

Comment 4. Line 199: the depth of penetration depends on the frequency.

Response: As per the suggestion, we have specified that deeper penetration of ultrasound is possible with employment of lower ultrasound frequencies.

[Line 224] “The advantage of ultrasound application for drug delivery to particular tissues and disease sites is its ability to be tightly focused, and using lower operating frequencies, potentially deeply inside the body [53].”

Comment 5. Line 201:  what range of ultrasound parameters will generate hyperthermia?  I think this will be a helpful piece of information to include for the three different complementary effects of ultrasound.

Response: Thank you for the suggestion. This is included in Section 1.6.1 “Thermal Effects of Ultrasound” and is in the low MHz range (0.5 to 7MHz). In Line 211 we now refer the reader to this section.

Comment 6. Line 229: This is not correct. Ultrasound gets absorbed by the tissue depending on the acoustic parameters and tissue properties. If ultrasound is not absorbed how the thermal effect is justified? In section 1.6.1 the authors are stating that the acoustic energy is absorbed.

Response: Thank you for pointing this out. This was a comment on the relative absorption of light and US energy in water. We understand that some tissues absorb more energy than others and therefore there will be relatively more heating in those tissues. We have changed the sentence to clarify its meaning.

[Line 240] “In comparison with light, ultrasound energy is absorbed relatively less within water and many soft-tissues, thus allowing it to penetrate deeply into the body and transmit energy to a precise location within target tissue [62].”

Comment 7. Line 336: what do the authors mean by beyond diffusion? This is not clear to me?

Response: We agree this was unclear and the manuscript has been modified accordingly.

[Line 303] “Depending on the parameters of ultrasound employed, within exposed tissues a given proportion of acoustic energy will be transformed into thermal energy, thus exerting thermal effects (Figure 3).”

Comment 8. It would be helpful for the readers if authors report the corresponding acoustic pressure (i.e., peak negative and peak positive) wherever they mention the intensity. E.g. line 360 what does 100 W/cm^2 correspond to in terms of peak rare factional and peak positive.

Response: This is not a straightforward conversion throughout, because the pressure at the ultrasound focus depends on the density of the particular tissue and the symmetry of the pressure wave, hence we have decided not to make this conversion throughout the article. However, for the original line 360 (Line 367) where the intensity of 1000 W/cm2 is mentioned, we have added the estimated peak negative pressure assuming the tissue is water and the pressure wave symmetrical = approximately 4 MPa.

Comment 9.  Line 386: what frequency?

Response: We have included the frequency of ultrasound applied in the study mentioned.

[Line 393] “Maruo et al. [12] showed that 55.5 kHz FUS, at an intensity of 50 mW/cm2 (acoustic pressure approximately 40 kPa) for 3 seconds, induced significant vasorelaxation in isolated canine arterial segments, with and without intact endothelium (precontracted with norepinephrine).”

Comment 10. Line 438: Does inertia cavitation cause transient or permanent permeability? AND
Comment 11. Line 439: Please explain how theoretically it is expected that stable oscillation of bubbles cause cell damage?  Are you referring to the generated shear stresses? Then mention the range.

Response: Thank you for your questions – it really depends on the magnitude of the effect. Large inertial cavitational activity may likely permanently increase the permeability through the BBB, if it induced significant tissue insult to the BBB. The original sentence added unnecessary confusion, so we have clarified it [Line 446]:

“It is generally believed that inertial cavitation events result in adverse, and potentially permanent, structural alterations to the BBB [88].”

Comment 12. Line 537: Phase-shift nano and micro droplets, which were introduced in 2000, have shown much enhanced stability and payload retention before reaching to the target. There are several papers on using phase-shift droplets for drug delivery. I think discussing the advantages of nanodroplets to liposomes and nanobubbles in terms of both stability and acoustic responsiveness would improve this review further.

Response: Thank you for the comment. We added this to the document as recommended, at line 746.

Comment 13. Line 580: why is using an unfocused stimulation beneficial?

Response: We were not implying that unfocused stimulation is necessarily beneficial, rather pointing out that unfocused ultrasound transducers were predominantly employed in these studies.

Comment 14. Line 625: What range of sizes are liposome and micelles are typically fabricated and studied? I did not find this information throughout the paper.

Response: The sizes of different NPs are detailed in Figure 6. To make the manuscript clearer, we also added this to the text at line 609.

Comment 15. Line 710: Can the authors explain how the formulation of lipid bilayer can trigger either thermal or mechanical effects of ultrasound?

Response: Thank you for the comment. The manuscript has been modified as recommended.

[Line 661] “In the case of liposomes, the lipid composition and physical characteristics play a prominent role in the sensitivity to the ultrasound [157, 185]. The membrane composition can be chemically modified to increase the sensitivity to ultrasound, as comprehensively discussed by Schroeder, A. et al. [14]. For example, the presence of amphiphiles, such as phospholipids with unsaturated acyl chains, increases liposome susceptibility to LFUS through destabilizing the lipid bilayer. Also, responsivity of liposome to ultrasound is greatly affected by introducing PEG-lipopolymers to the bilayer, most likely due to absorption of ultrasonic energy by the highly hydrated PEG head groups [14].”

Reviewer 4 Report

The authors have composed a comprehensive review on the novel therapeutic application of ultrasound to overcome blood-brain barrier in both cancerous and non-cancerous brain conditions and to facilitate drug release from nanoparticles. The review is very well written and organised, and is sufficiently detailed. I have no recommended changes.

Author Response

Comments and Suggestions for Authors: The authors have composed a comprehensive review on the novel therapeutic application of ultrasound to overcome blood-brain barrier in both cancerous and non-cancerous brain conditions and to facilitate drug release from nanoparticles. The review is very well written and organised, and is sufficiently detailed. I have no recommended changes.

Response:  Thank you for your time and consideration.

Round 2

Reviewer 3 Report

The authors have revised the manuscript adequately.